# Interdisciplinarity in Graduate Education for Groundwater Science and Technology

**Lu Xia** [1,*] **and Guohua Peng** [2]

1    School of Water Resources and Environment, China University of Geosciences (Beijing), Beijing 100083, China
2    Graduate School, China University of Geosciences (Beijing), Beijing 100083, China; pgh@cugb.edu.cn
*    Correspondence: xialu@cugb.edu.cn

**Abstract:** Groundwater science and technology is among the most rapidly developing branches of earth science globally. Interdisciplinarity poses both a challenge and a historical mission for graduate education in groundwater science and technology. This paper first analyzes the characteristics of domestic and international graduate education in groundwater science and technology. In addition, taking the China University of Geosciences (Beijing) as an example, it shows the history and development of the field in China. The results indicate that: (1) the graduate courses in groundwater science and technology in China are based on the characteristics of geoscientific research and closely integrate the advantages in environmental studies, ecology, and computer science to promote cross-fertilization across disciplines, departments, and universities; (2) after a few twists and turns in conferring master's and PhD degrees and in the construction of the discipline, groundwater science and technology has witnessed an increase in faculty members, expansion of the field of study, and the addition of modern educational and research facilities; (3) an increasing number of graduate students conduct research into the quality and safety of water supplies, rehabilitation technology of polluted water bodies, environment protection of river basin ecosystems, and so on.

**Keywords:** graduate education; groundwater science and technology; interdisciplinarity; innovation

## 1. Introduction

Groundwater science and technology is a branch of science based on geological theories [1]. It studies the structure and composition of the groundwater system, the spatial and temporal changes in groundwater quantity and quality due to the influence of human activities, and their interaction with the lithosphere, hydrosphere, atmosphere, and biosphere. It is an important subject of study for improving human life, meeting social needs, and fostering development in science and technology. Groundwater science and technology is among the most rapidly developing branches of earth science globally [2]. It has become the core and foundation of fluid earth science and plays an important role in solving various problems such as natural resource shortages, pollution, geological disasters, environmental degradation, climate change, hazardous waste, and treatment of greenhouse gases [3,4].

First introduced in Europe in the early 19th century [5,6], groundwater science and technology was then known as hydrogeology. Its main subject of study entailed describing and investigating the natural phenomenon of groundwater. In the 1930s and 1940s, a more complete, systematic, and independent scientific discipline started to be formed [7], focused on understanding the process of the formation and basic laws of groundwater. It was applied to the exploration and utilization of groundwater. Following the cross-fertilization of groundwater science and technology with other branches of science and advanced technological tools (e.g., mathematics, physics, chemistry, computer science, remote sensing, and information technology) [8], it gradually developed into a comprehensive discipline, existing at the margins of multiple disciplines. The nature of this field also transformed from basic science to applied basic and applied sciences.



Groundwater science and technology is a dynamic and cutting-edge field of study [9], inseparable from advanced scientific research and high-level graduate education. The core of graduate education in groundwater science and technology is to produce knowledge through systematic, specialized, and in-depth scientific research; to realize this, the key is to cultivate students' creativity. Interdisciplinarity is an inherent and prominent feature of groundwater science and technology. In the face of numerous practical problems related to groundwater, the discipline has become increasingly complex, multidisciplinary, horizontally extended, and multidimensional [10]. Therefore, in graduate education in groundwater science and technology, interdisciplinarity is inevitable. However, most departments in universities are set up according to the traditional demarcation of disciplines. In this context, universities are facing the problem of cultivating experts who can adapt to the calls for innovation in groundwater science and technology and promote graduate education across disciplinary boundaries.

This study analyzes the history and development of domestic and international graduate education in groundwater science and technology. It uses the graduate education in groundwater science and technology at the China University of Geosciences (Beijing) (CUGB) as an example. Therefore, the research objective of this context is the interdisciplinary model of graduate education in groundwater science and technology from three perspectives: designing courses, promoting research, and cultivating innovation. Additionally, we use the model of forming mentor groups to push for breakthroughs in interdisciplinary innovation ability, deepen the reform of the interdisciplinary model of graduate education, and cultivate high-level talents with innovation ability in the field of groundwater science and technology.

## 2. Materials and Methods

Groundwater science is a multidisciplinary field of study that plays an important role in solving various problems faced by humans today, such as pollution [11], geological disasters, and environmental degradation. Interdisciplinarity for graduate education in groundwater science and technology is a challenge. In order to solve this problem, research into graduate education in groundwater science and technology was conducted.

This paper collected research data on graduate education in groundwater science at CUGB from 1978 to 2021, including its enrollment scale, its courses, and its theses. The development of groundwater science and technology is studied by means of time series change, comparative analysis, and mathematical statistics. Firstly, it analyzes the characteristics of domestic and international graduate education in groundwater science and technology. Secondly, graduate education in groundwater science and technology at the China University of Geosciences (Beijing) is taken as an example to show the history and development of it in China from three perspectives—designing courses, promoting research, and cultivating innovation—in order to explore an interdisciplinary model of graduate education in groundwater science and technology. The variation in graduate enrollment scale with time is presented, and the increase in graduate courses on a 10-year basis is realized. Then, the research direction of groundwater science is divided into 12 categories, so the number of graduate theses in different research directions on a 5-year basis is counted for understanding the interdisciplinarity in groundwater science. Last, the model of a mentor group is used as the theoretical foundation to push for breakthroughs in the interdisciplinary innovation ability of graduate education in groundwater science and technology.

## 3. Results

### 3.1. Status of Graduate Education in Groundwater Science and Technology

3.1.1. Status of Graduate Education in Groundwater Science and Technology

Education in groundwater science and technology in words tends to distinguish emphases at the undergraduate and graduate levels [12–15]. In programs for undergraduate students, "hydrogeology" or "groundwater science and technology" do not exist as an

independent major. For graduate students, especially doctoral students, hydrology or groundwater science and technology is usually offered as a research direction in the department of geology or earth science. Many of these graduate students major in "geology" or "earth science" in their undergraduate studies, and some major in "civil engineering", "environmental science", "geography", and other science and engineering majors. Undergraduate education in the US stresses providing students with broad-ranging basic knowledge, whereas graduate education is more specialized and in-depth [16]. Therefore, courses related to groundwater are only offered in graduate schools for systematic studies. The basic requirement for doctoral students is to become high-level experts capable of independently carrying out creative research in the field.

3.1.2. History of Graduate Education in Groundwater Science and Technology in China

In the early years of the People's Republic of China, large-scale infrastructure construction, especially of mining and hydraulic facilities and large-scale industrial enterprises, necessitated hydrogeological and engineering geological surveys. At the time, professional education and the cultivation of experts in this field were absent in the country. Compared with the level of development in other parts of the world in that period, China was lagging by 20 years. In 1952, the Beijing Institute of Geology (renamed the China University of Geosciences (Beijing) in 1987) and Northeastern Institute of Geology (amalgamated into Jilin University in 2000) established the Department of Hydrogeology and Engineering Geology, officially starting undergraduate education in these disciplines in China. In 1954, graduate education in groundwater science and technology was initiated at the Beijing Institute of Geosciences with the help of Soviet experts. In the early stage of the development of graduate education in groundwater science and technology, there was a lack of faculty members, textbooks for graduate studies, and hardware supplies [17]. The period before 1978 was one of exploration in graduate education in groundwater science and technology in China, which was based on the Soviet model. In terms of the mode of training, coursework mainly depended on students' self-learning; furthermore, teaching, research, and practice were closely integrated with engineering needs. There was no standardized curriculum or plan. The textbooks used initially were translated from the Soviet Union; later, officially compiled textbooks were published. In 1978, related universities began accepting applications for master's or doctoral studies in groundwater science and technology. In the 1980s, Chinese scholars began to pay attention to the trends in groundwater science and technology education in countries such as the US and Canada, as well as in Europe, borrowing from the experience of other more advanced countries and gradually improving the compilation of textbooks domestically. In 2011, the Catalog of Academic Disciplines and Majors in Graduate Education in China was revised for the fourth time, clarifying the name of the discipline as "hydrogeology", placed under the first-level scientific discipline "geology" as a science to study groundwater (hydrosphere). Guided by theories of earth system science, with the physical, chemical, and biological effects of water and rock (soil) at the core, groundwater science and technology investigates the formation and evolution of groundwater under the influence of nature and human beings, as well as its impact on resources and the environment through interactions with the mantle, lithosphere, biosphere, and atmosphere. Groundwater science and technology provides scientific support for the rational exploration and utilization of groundwater resources to achieve harmony between humans and nature. Upgrading the construction of the discipline, adjusting the research directions scientifically, promoting the reform of graduate education in hydrogeology, and improving the quality of the cultivation of graduate students in hydrogeology, groundwater science and technology features the cultivation of talents and interdisciplinarity. Over the past 60 years, graduate education in groundwater science and technology in China has developed rapidly, forming an interdisciplinary system in the training of graduate students.

### 3.1.3. Development of Graduate Education in Groundwater Science and Technology at CUGB

From 1978 to 2000, the size of the body of graduate students in groundwater science and technology remained relatively stable. After the modification of the Catalog of Academic Disciplines and Majors in Graduate Education in China in 1997, the discipline of hydrogeology was canceled. CUGB continued to train graduate students in groundwater science and technology under hydrology, a second-level discipline of "hydraulic engineering." Later, the discipline was termed groundwater science and technology, a second-level discipline of "geological resources and geological engineering", under which graduate students were trained. During this period of adjustment, the number of students decreased, which impacted the development of the discipline of hydrogeology and graduate education. In 2014, after a fourth round of adjustment, the recruitment of doctoral and master's students in hydrogeology resumed under the first-level discipline "geology" to train graduate students oriented toward academic research; subsequently, the number of graduate students in hydrogeology greatly increased. Figures 1 and 2 show the number of master's and doctoral students in groundwater science and technology at CUGB over the past 40 years. Graduate education in groundwater science and technology in China has experienced a process of inception, development, and gradual improvement. After a few twists and turns in conferring master's and PhD degrees and in the construction of the discipline, groundwater science and technology has witnessed an increase in faculty members, expansion of the field of study, and the addition of modern educational and research facilities. With a more intensive integration of industry, teaching, and research, a more comprehensive system of graduate education has been formed.

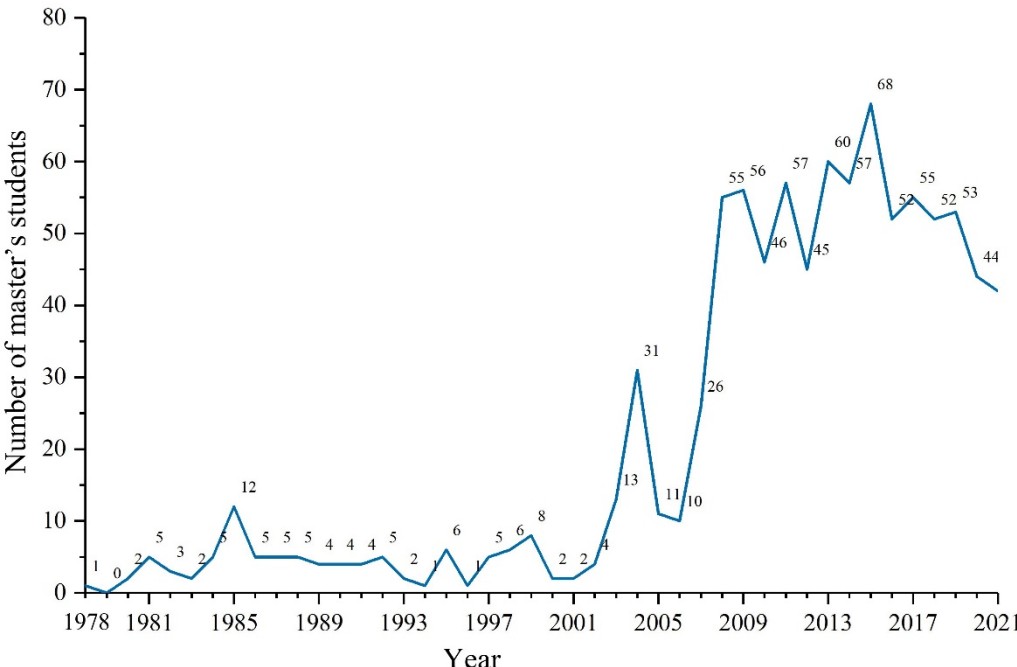

**Figure 1.** Number of master's students in groundwater science and technology at China University of Geosciences (Beijing), 1978–2021.

CUGB is a research-oriented university with geology, natural resources, and environment study as its principal fields. The six disciplines of geosciences, engineering science, environment/ecology, materials science, chemistry, and computer science are in the top 1% of the essential science indicators (ESI), with geosciences being in the top 1‰ [18]. Against this background, the discipline of groundwater science and technology is firmly based on the characteristics of geoscientific research and closely integrates the advantages in environmental studies, ecology, and computer science to promote cross-fertilization across

disciplines, departments, and universities, as well as with the entire industry. A research team with capacities in extensive interdisciplinary cross-fertilization has been formed to identify the right scientific problems and solve contemporary dilemmas.

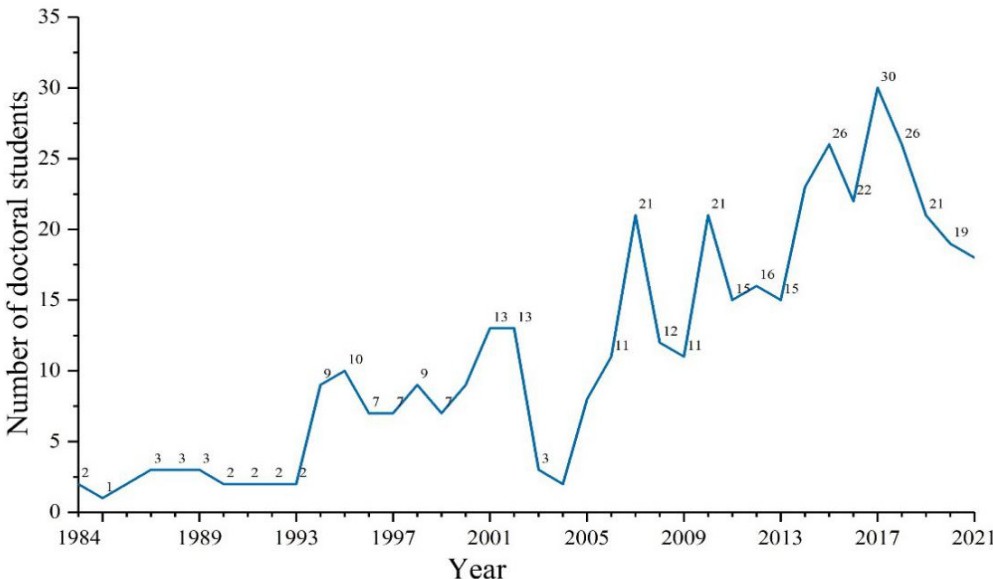

**Figure 2.** Number of doctoral students in groundwater science and technology at China University of Geosciences (Beijing), 1984–2021.

### 3.2. Interdisciplinarity in Graduate Education in Groundwater Science and Technology

The scope of groundwater science and technology research has turned from solving practical and concrete problems of production to the long-term harmonious development of the relationship between humans and nature. In graduate education in groundwater science and technology, the trend of integrating elements of multiple disciplines has emerged.

#### 3.2.1. Designing Courses for Graduate Education

In terms of offering specialized courses in graduate education in groundwater science and technology, the number of courses has been gradually refined and increased, from three basic ones (general hydrogeology, advanced hydrogeology, and groundwater dynamics) in the 1950s and 1960s, to nearly 20 specialized courses with different research directions and disciplinary elements. In the 1970s and 1980s, the development of basic theories in groundwater science and technology was rapid, precipitating the transition from qualitative development to quantitative research. The graduate curriculum focused on basic hydrogeological theories and methods. Courses included numerical analysis of groundwater movement, mathematical programming for groundwater resources management, hydrodynamics, hydrogeology of bedrock fissure water, mathematical geology, and so on. In the 1990s, groundwater science and technology gradually transformed from traditional hydrogeology to modern hydrogeology. New technologies, such as computer and remote sensing, and new methods, such as fractal theory, information theory, and system theory, have been fully introduced into the discipline of hydrogeology. Gradually, graduate courses were enriched, showing initial signs of interdisciplinary integration. Examples included the groundwater information system, water resources evaluation and management, finite element programming, random hydrology, groundwater environmental engineering, methods in environmental systems engineering, environmental water quality models, applications of computer science in water engineering and environment, and geological quality standards in turbidity.

In the past 20 years, hydrogeology has incorporated modern technologies and is more intimately integrated with other disciplines, gradually forming a comprehensive and interdisciplinary field. The graduate course offerings have become more extensive,

now including Chinese regional hydrogeology, the groundwater information system, water resources evaluation and management, groundwater pollution and control, environmental geochemistry, soil hydrodynamics, random methods in hydrogeology, theory of regional groundwater flow, groundwater simulation technology, remote sensing for water environment monitoring, water resources planning and management, engineering fluid mechanics, modern hydrological simulation and forecasting, development in water resources and environment, environmental ecology, and urban geology and environment. Interdisciplinarity has become a necessity in the development of the discipline today [19]. To analyze and deal with problems of the groundwater system, geochemistry, biology, geotechnics, soil science, remediation and treatment techniques, and other related knowledge are indispensable. Therefore, it is important to offer relevant courses in a timely manner to promote the development of graduate education in hydrogeology.

To this end, a "course cluster" [20] system of graduate courses has been formed to develop an interdisciplinary approach to graduate education. The groundwater science and technology curriculum is formulated based on the cross-fertilization of disciplines, highlighting the characteristics of advantageous disciplines of the university, considering the basic skills of each research direction, and integrating them into the "course cluster" to form a logical and progressive course system. It is also formulated to explore the establishment of a new mechanism of training across departments, disciplines, and majors. The courses offered in the "course cluster" can be named after the individual industry, with the person in charge of major national projects in industry acting as the course leader and organizing the project team according to the characteristics of interdisciplinarity in industry.

The persons in charge of subprojects are responsible for teaching the knowledge of the related disciplines. For example, the industry of groundwater pollution remediation entails knowledge in many fields such as soil, the environment, geological disasters, and ecology. "Groundwater pollution remediation" can be established as a course in the course cluster. The chief scientist who has undertaken such projects for many years can be invited to lead this course, and the persons in charge of subprojects can teach the knowledge of related disciplines. The project team has both an interdisciplinary research paradigm and experience in multidisciplinary collaborative research for complex scientific and technological problems. The curriculum is oriented to major scientific problems and is characterized by interdisciplinarity, thus forming a new system of courses in groundwater science and technology.

### 3.2.2. Interdisciplinarity in Graduate Students' Research Work

The graduate student thesis is the main manifestation of the quality of education and level of academic achievement [21]. It is an important reflection of graduate students' innovation ability and the historical evidence of their research work. The authors counted 958 master's and doctoral theses in the discipline of hydrogeology at CUGB from 1978 to 2021 and categorized each thesis by research direction. Figure 3 shows the results. The theses were divided into 12 research directions [22]: fundamentals of hydrogeology, groundwater dynamics (groundwater hydraulics), hydrogeochemistry, groundwater microbiology, hydrogeothermics, subsurface hydrosphere history (paleohydrogeology), hydrogeological methodology, exploration hydrogeology, regional hydrogeology, groundwater disaster prevention (engineering hydrogeology), subsurface hydrosphere protection (environmental hydrogeology), and hydrogeoecology.

Many of the above 12 research directions have been developed by integrating the knowledge of other disciplines such as hydrogeochemistry, groundwater microbiology, hydrogeothermics, engineering hydrogeology, environmental hydrogeology, and hydrogeoecology. Hydrogeochemistry mainly researches the chemical elements of groundwater and the law of their migration in water, which needs to integrate the knowledge of geology, hydrology, and geochemistry. Groundwater microbiology studies microorganisms in the groundwater and their interaction, and extensive knowledge of the microorganisms is required. Environmental hydrogeology mainly considers groundwater pollution, resource

depletion, water quality, human health, environmental restoration, legal system, etc. This research direction has brought about a new interdisciplinary, applied, and marginal discipline [23]. It is not only closely related to geology, hydrology, and environmental sciences but also to the knowledge of natural science, such as physics, chemistry, meteorology, agriculture, water conservancy, and social sciences [24].

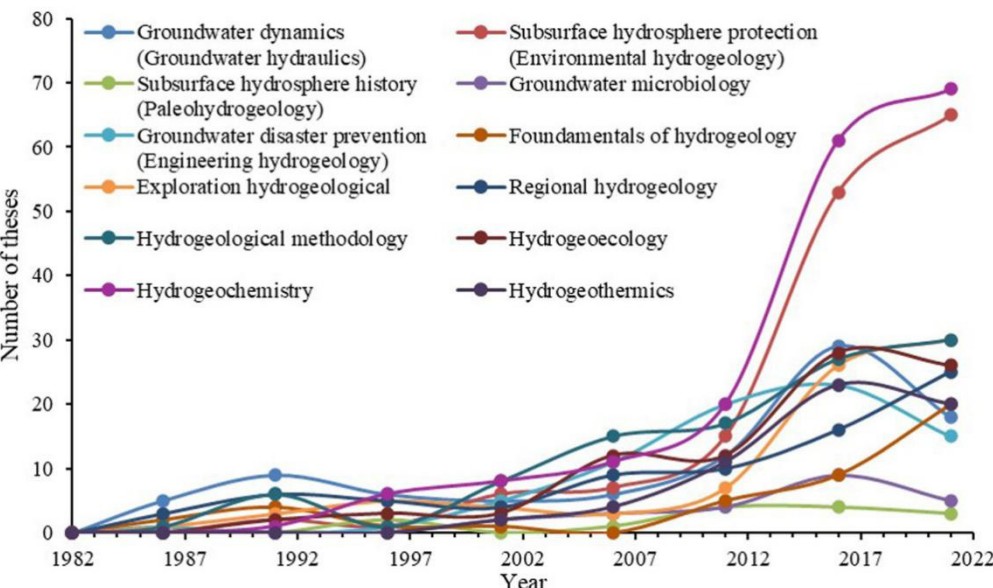

**Figure 3.** Categorization of graduate students' theses in groundwater science and technology at China University of Geosciences (Beijing), 1978–2021.

As Figure 3 shows, during the transitional period from traditional hydrogeology to modern hydrogeology, a relatively comprehensive and systematic disciplinary system in groundwater science and technology was formed. The practical and research activities of graduate students mainly focused on research into the formation process and basic laws of groundwater. Groundwater dynamics and regional hydrogeology were the two main research directions in the theses. In the 1990s, advanced technologies and methods from computer science were gradually incorporated into groundwater science and technology, and the quantitative method was effectively applied and incorporated into the evaluation, estimation, and management of water resources. Research into groundwater resources witnessed the transition from traditional research methods to modeling, and the discipline of hydrogeology was more closely integrated with other disciplines. With the rapid development of society and economic growth, environmental problems and geological disasters caused by the overexploitation of groundwater resources through human activities in urban areas and by industrial and agricultural production have gradually emerged. Graduate students' research directions have, thus, also turned to environmental hydrogeology, hydrogeochemistry, and hydrogeological methodology. Since the beginning of the 21st century, the severity of the environmental and disastrous impact related to groundwater has further increased. The depletion of natural resources and deterioration of the ecosystem have become serious threats to the security and development of human beings. It is, therefore, urgent to build hydrological systems, environmental systems, and ecosystems where humans and nature can coexist and create virtuous cycles. As such, the research directions of environmental hydrogeology, hydrogeochemistry, and hydrogeoecology have become the most popular choices. Furthermore, an increasing number of graduate students conduct research into the quality and safety of water supplies, rehabilitation of polluted water bodies, environment protection of river basin ecosystems, geological capture of carbon dioxide, geological treatment of nuclear waste, and development and utilization of geothermal resources. The content of graduate students' research and practical work displays interdisciplinarity in the research and treatment of problems.

*3.3. Breakthroughs in Interdisciplinary Innovation*

Interdisciplinary innovation provides support for emerging sciences. Groundwater science and technology is an interdisciplinary and comprehensive discipline. For universities, in the process of cultivating graduate students' innovation ability, in addition to offering a systematic, diverse, and high-quality graduate curriculum and practical training to improve the ability of technical work in their field of expertise through coursework but also develop the positive potential of graduate students in accordance with the laws of research and development of graduate students.

Universities can set up a special plan for interdisciplinary graduate training. The plan forms mentor groups that have at least two members and experience in major national projects. The main and deputy supervisors come from different colleges and different disciplines. In addition, it adopts the enrollment mode of mentor groups in this plan to make them work together to train graduate students. Moreover, the plan can construct groups of graduate students with different research backgrounds in scientific research activities to collaborate on basic scientific problems in groundwater science and technology or applied technologies in frontier fields with the assistance of an interdisciplinary supervisory faculty. Graduate education should try to nurture positive experiences for innovation among graduate students, establish their confidence and initiative in innovation, cultivate their positive mindset toward innovation, and use positive assessment in scientific research work as an evaluation standard to enhance their innovation ability and achieve breakthroughs in interdisciplinary innovation.

## 4. Conclusions

(1) The graduate courses of groundwater science in China are based on the characteristics of geoscientific research and closely integrate the advantages in environmental studies, ecology, and computer science to promote cross-fertilization across disciplines, departments, and universities, as well as the entire industry. (2) Advanced technologies and methods are integrated into the scientific research of groundwater science, as well as elements of other disciplines. The content of graduate students' research and practical work displays interdisciplinarity in the research and treatment of problems. (3) After a few twists and turns in conferring master's and PhD degrees and in the construction of the discipline, groundwater science has witnessed an increase in faculty members, expansion of the field of study, and the addition of modern educational and research facilities. (4) An increasing number of graduate students conduct research into the quality and safety of water supplies, rehabilitation of polluted water bodies, environment protection of river basin ecosystems, geological capture of carbon dioxide, geological treatment of nuclear waste, and development and utilization of geothermal resources.

The core of graduate education is to create knowledge through systematic, specialized, and in-depth scientific research, with a focus on production practice and innovation consciousness. Close integration with the frontiers of the world's scientific and technological development to cultivate creative scientific thinking is the source of interdisciplinarity. The cross-fertilization and mutual infiltration between different disciplines has become an urgent need of current socioeconomic development and an inevitable trend of scientific development. Coordinating and deploying interdisciplinary research oriented toward major national strategic needs and emerging scientific frontiers can help promote multidisciplinary collaborative research into complex scientific and technological problems and form new research directions in the discipline. Interdisciplinarity is indispensable in graduate education, and it poses both a challenge and a historical mission for graduate education. The scope of groundwater science research has turned from solving practical and concrete problems of production to the long-term harmonious development of the relationship between humans and nature. In graduate education in groundwater science, the trend of integrating elements of multiple disciplines has emerged. There are many opportunities for graduate education. Moreover, innovative concepts can be introduced early into education by interdisciplinarity in graduate education for groundwater science

and technology and trigger young graduates' interest in achieving sustainable development. By scientifically optimizing the system of courses, offering a comprehensive and multidisciplinary curriculum, and forming mentor groups, promoting an interdisciplinary model of education can cultivate first-grade experts in groundwater science and technology with a multidisciplinary background and capacities in application, interdisciplinary talents, and technical skills. This can enable them to pursue breakthroughs in interdisciplinary innovation ability in groundwater science and technology and achieve technical innovations that combine theories of groundwater science and technology with practical applications in geological resources and the environment.

**Author Contributions:** Conceptualization, L.X. and G.P.; methodology, L.X.; formal analysis, G.P.; investigation, L.X.; resources, G.P.; data curation, L.X.; writing—original draft preparation, L.X.; writing—review and editing, G.P.; funding acquisition, L.X. All authors have read and agreed to the published version of the manuscript.

**Funding:** This research was funded by the Chinese Society of Academic Degrees and Graduate Education, grant number "2020MSA174" and the Beijing Association of Higher Education, grant number "YB202114".

**Institutional Review Board Statement:** Not applicable.

**Informed Consent Statement:** Not applicable.

**Data Availability Statement:** The data that support the findings of this study are available from the corresponding author.

**Acknowledgments:** The authors would like to thank the reviewers for their valuable comments and Shupei Wu for her assistance.

**Conflicts of Interest:** The authors declare no conflict of interest.

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
