# Peer review of "Interdisciplinarity in Graduate Education for Groundwater Science and Technology"

_sustainability, doi:10.3390/su14095645_

Round 1
Reviewer 1 Report
The article is based on a historical and practical analysis of the development of the discipline "Groundwater Science and Technology", and an extension to a multidisciplinary conceptual framework. By being implemented in three parameters: designing courses, promoting research, and cultivating innovation.
The authors were able to review in the qualitative (descriptive) approach and the quantitative approach, through the three parameters, about the development in the discipline "Groundwater Science and Technology".
At the same time, the use of " Graduate Education….", in my opinion, invites an important element, which has disappeared or been mentioned in the margins, and that is community education, or connecting and intertwining the (new) multidisciplinary with the formal education system (pupils and young people).
The authors used two sources: [8] and [18] to connect the conceptual framework they developed both to other disciplines, and to the community or urban humanity. Unfortunately, the connection of development (which the whole article revolves around) and its products, ignored the educational elements that must be assimilated in the community or education system (pupils of today - future leaders) to improve the connection between humanity and natural resources and the nature itself.
Therefore, this argument brings me back to the authors' division into 12 content categories of theses, which also in this division ignored the educational and community element (there might not have been theses in this area. If so, they should have noted this).
Reviewer 2 Report
Dear authors,
Congratulations on the analyzed topic. In the current context of environmental concerns, the analysis launched by you is of interest. However, I would suggest that you focus more on the implications of training more or fewer specialists in this field on environmental issues, and the impact that young graduates have on the economy and society. It is also very important to make a connection with the Sustainable Development Agenda and the importance of higher education in achieving the goals.
Best regards,
Gabriela Neagu
Reviewer 3 Report
The research is interesting as a case study, insofar as conclusions can be extrapolated for this area of training, beyond the case of education in China.
In subchapter 2 further clarifications are needed: what type of research data was collected from 1978-2021, what is the research problem, what is the purpose of the research. The research objectives should be explicitly named (even if they exist in the text). How the comparative analysis was performed, how the statistical analysis was used.
The use of the forming mentor group is announced, but the model is not sufficiently explicit and argued.
The conclusions are too general, they correlate to little with the objectives of the study (perhaps precisely because they are not clearly formulated)
Round 2
Reviewer 1 Report
Dear Authors
Thanks for the upgrade in the issues I raised.
Sincerelyת
Reviewer 3 Report
Thank you for the improved version of your article. We appreciate the whole effort. We consider that the following aspects have not been enough clarified (perhaps we did not notice them in the text, if they are please just underline them, for viewing): research problem, the purpose, the research objectives.
